# Discovering Creative Behaviors through DUPLEX: Diverse Universal Features for Policy Exploration

**Borja G. Leon**[*]
Iconic
borja@iconicgames.ai

**Francesco Riccio**
Sony AI
francesco.riccio@sony.com

**Kaushik Subramanian**
Sony AI
kaushik.subramanian@sony.com

**Peter R. Wurman**
Sony AI
peter.wurman@sony.com

**Peter Stone**
Sony AI
The University of Texas at Austin
peter.stone@sony.com

## Abstract

The ability to approach the same problem from different angles is a cornerstone of human intelligence that leads to robust solutions and effective adaptation to problem variations. In contrast, current RL methodologies tend to lead to policies that settle on a single solution to a given problem, making them brittle to problem variations. Replicating human flexibility in reinforcement learning agents is the challenge that we explore in this work. We tackle this challenge by extending state-of-the-art approaches to introduce DUPLEX, a method that explicitly defines a diversity objective with constraints and makes robust estimates of policies' expected behavior through successor features. The trained agents can (i) learn a diverse set of near-optimal policies in complex highly-dynamic environments and (ii) exhibit competitive and diverse skills in out-of-distribution (OOD) contexts. Empirical results indicate that DUPLEX improves over previous methods and successfully learns competitive driving styles in a hyper-realistic simulator (i.e., GranTurismo™ 7) as well as diverse and effective policies in several multi-context robotics MuJoCo simulations with OOD gravity forces and height limits. To the best of our knowledge, our method is the first to achieve diverse solutions in complex driving simulators and OOD robotic contexts. DUPLEX agents demonstrating diverse behaviors can be found at https://ai.sony/publications/Discovering-Creative-Behaviors-through-DUPLEX-Diverse-Universal-Features-for-Policy-Exploration/.

## 1 Introduction

In non-stationary complex environments, reinforcement learning (RL) [1] agents are compelled to exhibit flexible and diverse behaviors to robustly adapt to different scenarios and interact with other actors [2, 3]. To this end, a growing community is researching methodologies to train agents that, unlike conventional RL [4, 5], can solve tasks with a diverse set of near-optimal strategies [6]. Such methodologies are explicitly crafted to enhance the exploration of the state-action space, equipping

---

[*]Work done at Sony AI.

38th Conference on Neural Information Processing Systems (NeurIPS 2024).

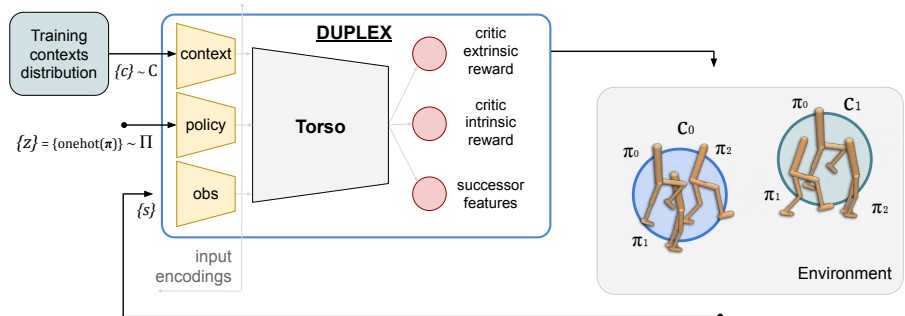

Figure 1: DUPLEX data flow while training three policies over a set of two contexts. At each iteration, we provide three inputs to our multi-policy agent: (i) a context vector describing task requirements and environment dynamics in the current episode $c$; (ii) an encoding of the policy used in the episode $z$; (iii) and the current state of the environment $s$. The critic network returns estimates for the intrinsic and extrinsic rewards and successor features to drive diverse behavior discovery. Finally, the algorithm samples policies in $\Pi$ uniformly and rolls them out to collect more experience.

agents with the capability to discover qualitatively diverse solutions for a given task distribution. For example, when planning a route from home to work, a human commuter might identify one route that uses the highway and one that sticks to side streets. While using the highway may be optimal in expectation, using the alternative route may be called for different contexts that influence the traffic report, or weather forecast.

Existing approaches struggle to generalize to highly dynamic environments and are designed to learn in single-task settings [3, 7]. To alleviate this, we introduce *Diverse Universal features for PoLicy EXploration* (DUPLEX), an algorithm that trains an agent to optimize a set of diverse policies over different contexts. In this work, contexts comprise two components: a task requirement and a description of the current environment dynamics. To better ground such concepts, let us refer to the previous commuter example. In this case, we would refer to task requirements as time-to-completion. On the other hand, factors such as the weather forecast and traffic report would form part of the environment dynamics DUPLEX builds on top of the strengths of diversity learning [6] in RL and universal estimators (UE) [8, 9] to introduce a novel methodology that preserves both performance and diversity in highly dynamic environments and multi-context settings – which is key to enabling human-level task execution. For example, if the commuter is a human that injures their leg, they can immediately balance on one leg and even jump forward without using the injured leg. We aim to transfer this adaptability to agents in different contexts. To this end, we adopt the Contextual Markov Decision Process (CMDP) framework [10], where different episodes correspond to different contexts. As shown in Figure 1, DUPLEX is designed to receive three inputs representing the encoding of the context, policy, and state. Contexts are represented as feature vectors and uniformly sampled from a discrete set of predefined task requirements and dynamics settings (e.g., weather condition); the $i$-th policy in the set of policies $\Pi$ is selected and passed to the model (e.g., different routes); and the state comes directly from the environment observation. Then, the critic output is split into different heads to support the estimation of the extrinsic and intrinsic rewards, and successor features (SFs) [11]. We refer to the reward coming from the environment as the extrinsic reward, while the metric computed to promote diversity is the intrinsic reward. It is worth noting that, at the beginning of each training episode, we uniformly sample a context from $C$ and a policy from $\Pi$ to collect experience. DUPLEX then iteratively trains the set of policies to maximize both the distance in their successor features (guided by intrinsic rewards) and the task performance (given by the extrinsic rewards). We rely on SFs to measure distances as they, by definition, represent the state features that a given policy is expected to experience along a trajectory, and thus, they are an intuitive way to quantify policy diversity.

Hence, our main contribution is DUPLEX, a novel algorithm that contributes to diversity learning in RL by improving on previous work to better preserve the diversity vs. near-optimality trade-off in highly-dynamic environments and multi-context settings. We evaluate our approach on the real-time, physically realistic, car-racing simulator GranTurismo™ 7 (GT) [12] and on two multi-task MuJoCo environments with changing dynamics (Walker2D, Ant) [13]. Our experimental results indicate that

DUPLEX improves over previous state-of-the-art RL diversity baselines [6] and UE baselines [8, 9]. In fact, DUPLEX is the only algorithm to learn diverse competitive policies in GT and outperforms other baselines when evaluated in out-of training distribution (OOD) contexts. Moreover, we conduct a detailed ablation study that isolates the effects of each of DUPLEX's main components.

## 2 Related Work

In this work we build on top of two main bodies of research: diversity learning and universal function approximators. We organize related work accordingly.

**Diversity.** Diversity learning is gaining attention within the research community, due to the important benefits it yields to autonomous agents. Such benefits include generalization [14], exploration [15, 16], creativity [3], and self-play **(author?)** [2]. In this work, we extend the application of diversity learning to highly-dynamic environments and multi-context settings. A central precursor to our work is DOMiNO [6]. DOMiNO employs RL to maximize extrinsic and intrinsic diversity rewards, which are combined with Lagrange multipliers and Van der Waals (VdW) forces to balance diversity and performance. DOMiNO has two variants: one bases its diversity objective on the average of input features which grants stability at the cost of storing a moving average for every task – limiting scalability. The second variant uses the critic to estimate SFs, which solves the scalability issue at the cost of instability due to the added learned target. Intuitively, such limitations prevent DOMiNO from being effective at discovering diverse behaviors in CMDPs. Further, the Lagrange multipliers and VdW forces can limit diversity in complex environments where creative behaviors need significant exploration before providing satisfying returns [17]. Such limitations become more evident in highly dynamic environments such as GT (see Section 5). Our algorithm is instead designed to tackle these limitations.

Quality-Diversity (QD) is another active line of work within diversity learning that involves evolutionary methods. QD aims to generate large collections of diverse and high-performing solutions, primarily through evolutionary optimization [18, 19]. In QD, diversity is measured through a descriptor space defined by the user [20]. Some of the main approaches in QD include MAP-Elites [21, 22], local competition [23] and more recently those that combine RL with QD [24, 25, 26]. Closer to our research, **(author?)** [27] introduce a variant for MAP-Elites to learn diverse behaviors for multiple tasks within the training distribution. Our framework diverges from QD in several ways. First, QD algorithms emphasize diversity through evolutionary strategies, while our method relies on maximizing a reward objective. Second, QD requires the user to define a diversity descriptor, while we only restrict diversity to be in the near-optimal space. Third, one of our main goals is finding competitive diverse behaviors for OOD tasks and dynamics, an objective that has not been studied in Quality-Diversity (QD).

**Universal Function Approximators.** To make more robust estimations of the policies' expected behaviors, and thus to better quantify diversity, our work pivots around universal function approximator (UE). UE research is grounded on factoring the value estimates separating states from tasks [8] and policies [28]. Notably, **(author?)** [9] improve the formalization of successor features [11], an estimation of state-action visitation, by conditioning their estimation on both task and policies. Still, approaches based on SFs are brittle in complex domains [29, 30], requiring sophisticated network architectures [31, 32] to work effectively. Akin to **(author?)** [33], we adopt a similar approach to transferring learning of SFs to continuous domains by combining them with SAC [4]. However, we achieve an improved estimation of the expected SFs by employing the average of the critic outputs and by adding the entropy term to the SFs learning objective. We find that the addition of the entropy term is a novel component that DUPLEX carries with and it is key to improving robustness and performance (see Appendix B).

## 3 Background and Notation

We briefly introduce the main building blocks of DUPLEX and the notation we adopt. First, we review basic concepts of multi-task RL and explain how it can be represented by the contextual-MDP framework. Next, we describe how universal estimators are used to enhance generalization across contexts (especially when context enumeration is impractical).

**Multitask RL.** We consider training a multi-policy RL agent that solves a CMDP [10] represented as a tuple $\mathcal{M} = \langle \mathcal{S}, \mathcal{A}, P, R, \gamma, \mu_C, \mu_S \rangle$, where $\mathcal{S}, \mathcal{A}$ are the state, action spaces respectively, $R$ is the reward function, $P$ is the unknown transition function, $\mu_S$ is the initial state distribution conditioned on the context, and $\mu_C$ is the context distribution. We use context $c \equiv \{u, w\}$ to summarize both information about the particular dynamics $u$ of that environment (e.g., the effect of gravity), and information about the task $w$ in the current episode (e.g., position constraints). Every episode starts with a context $c \sim \mu_C$ and an initial state $s_0 \sim \mu_S(\cdot \mid c)$. Then, at each time step $t$, the agent selects an action $a_t$ according to its policy $\pi(\cdot \mid s_t, c)$, receives a reward $r_t \sim R(s_t, a_t, c)$ and transitions to the next state $s_{t+1} \sim P(\cdot \mid s_t, a_t, c)$. Policies are characterized by their state-action occupancy, i.e., how often a policy visits a state-action pair. We consider two state-action occupancy metrics: $d_{\pi_c}^{\mathrm{avg}}(s, a) = \lim_{T \to \infty} \frac{1}{T} \mathbb{E} \sum_{t=1}^{T} \mathbb{P}_{\pi_c}(s_t = s) \pi(s, a, c)$ for the average occupancy metric and $d_{\pi_c}^{\gamma}(s, a) = (1 - \gamma) \mathbb{E} \sum_{t=1}^{\infty} \gamma^t \mathbb{P}_{\pi_c}(s_t = s) \pi(s, a, c)$ for the discounted case – where $\mathbb{P}$ is the probability measure of states at $t$ induced by $\pi$ in $c$. The objective of our algorithm is to find a set of diverse policies that maximize the expected return in every context $\max_{d_{\pi_c} \in \mathcal{K}} \sum_{s,a,c} r(s, a, c) d_{\pi_c}(s, a)$, where $\mathcal{K}$ is the set of admissible distributions [34].

**Universal estimators.** When aiming to solve multiple tasks, we follow the common approach of decomposing the input of the neural network to facilitate transfer learning between tasks and policies. Methods that are relevant for our work include "Universal Value Function Approximators" (UVFA) [8], which add a task-descriptor vector $w$ as input to a value function approximator parameterized by $\theta$, $V_\theta(s, a, w)$. If $V_\theta$ is smooth w.r.t. $w$, then $V_\theta$ is expected to generalize across tasks within the training task space. Akin to previous works [11, 6], we assume that every state-action pair is correlated with observable *features* known as "cumulants" $\phi(s, a, c) \in \mathbb{R}^d$. These cumulants can either be given through relevant properties within the state[2] or be learned. In a similar fashion to value functions, we can define *expected features* $\psi_{\pi_c}(s, a) = \mathbb{E}_{s', a' \sim d_{\pi_c}(s,a)} \phi(s', a', c) \in \mathbb{R}^d$, which we will refer to as average ($\psi^{\mathrm{avg}}$) or discounted ($\psi^\gamma$) expected features when using $d_{\pi_c}^{\mathrm{avg}}$ and $d_{\pi_c}^{\gamma}$ respectively. In the discounted case, $\psi^\gamma$ are also known as "successor features" (SFs). This latter formalization is key to our work and also to USFA [9], a general framework that combines UVFA-like task decomposition with the SFs, and exploits generalized policy improvement algorithm (GPI) **(author?)** [11]. Importantly, USFA disentangles both tasks and policies by giving as input a vector $z$ that is a representation of the current policy $z = e(\pi)$, where $e$ is an encoding function – in our approach $e$ encodes policies as one-hot vectors $z$.

# 4 Learning Near-Optimal Diverse Behaviors in Multi-Goal Continuous Settings

DUPLEX is a method designed to provide diverse solutions to complex tasks by simultaneously maximizing both the true reward and dissimilarity within a set of policies $\Pi$ across the different environments of a CMDP. Thus, according to previous work [6] we define diversity as:

**Definition 4.1 (Diversity)** $\mathrm{Diversity}(\Pi)$ *is a metric of dissimilarity among policies in a set $\Pi$ with a common goal. Formally, if $\psi_{\pi_i}$ and $\psi_{\pi_j}$ are a function of state-occupancy of relevant features of any two policies in $\Pi$, then their dissimilarity is given by $||\psi_{\pi_i} - \psi_{\pi_j}||$. A non-zero value of this norm indicates dissimilarity, with larger values indicating greater divergence between the policies. Mathematically, diversity is defined as the sum of the minimum L2 dissimilarity norms in $\Pi$:*

$$\mathrm{Diversity}(\Pi) = \frac{1}{2\,size(\Pi)} \cdot \sum_{\substack{\forall \pi_i, \pi_j \in \Pi, \\ i != j}} \min ||\psi_{\pi_i} - \psi_{\pi_j}||_2^2$$

Using this definition, we formulate our learning problem and describe DUPLEX's main contributions in the following sections. We first extend the formulation of diversity learning to context-conditioned environments, then we introduce a novel mechanism to more efficiently control the diversity-performance trade-off, and finally, we describe how to improve SFs estimation.

---

[2]For example, in a car racing scenario features could correspond to relevant information for the car such as the course velocity or if a car has crashed into another

## 4.1 Context-conditioned Diversity Learning

Similar to **(author?)** [6], our objective is to maximize diversity within a set of policies $\Pi$. We express the distances between policies in $\Pi$ in terms of expected features $\psi$, which are a function of their state-occupancy ([9, 30]). We measure $\psi$ distances following Def. 4.1 and use a context-based Hausdorff distance [35] to enforce context-conditioned diversity within $\Pi$. We aim at training an RL agent that, given a context $c$, discovers a set of $n$ near-optimal policies $\Pi = \{\pi_c^i\}_{i=1}^n$ by successfully optimizing:

$$\max_{\Pi} \text{ Diversity}\,(\Pi) \text{ s.t } d_{\pi_c} \cdot r_e \geq \rho \hat{v}_e, \quad \forall \pi_c \in \Pi \tag{1}$$

where $d_{\pi_c}$ is the occupancy metric of a context-conditioned policy, $r_e$ is the environment reward (or *extrinsic* reward), $\rho \in [0, 1]$ is a hyper-parameter defining the near-optimality region that we refer to as *optimality-ratio*, and $\hat{v}_e$ is the value of a target policy. In practice, the target policy refers to a policy in $\Pi$ that ignores the diversity objective and is trained exclusively to maximize the extrinsic rewards. As in [6], the policy value $v$ expresses the expected reward accumulated by a policy $\pi$. According to Definition 4.1, diversity is a distance over a set of occupancy metrics Diversity : $\{\mathbb{R}^{|S||A||C|}\}^n \to \mathbb{R}$ and Eq. 1 is defined to maximize the diversity of the occupancy metrics and preserve the near-optimality of the policies in $\Pi$.

To promote diversity between different policies, we adopt the repulsive reward from [6] and extend it to be conditioned on the context. More formally, given the tuple $\langle s, a, \pi^i, c \rangle$ and a cumulant function $\phi$, the diversity reward that we maximize for is:

$$r_d^i(s, a, c) = \phi(s, a, c) \cdot (\psi_{\pi_c^i} - \psi_{\bar{\pi}_c^i}) \tag{2}$$

where $\psi_{\pi_c^i}$ are the expected features from policy $\pi^i \in \Pi$ at $(s, a, c)$ and $\bar{\pi}_c^i \in \Pi$ refers to the policy with the closest expected features to $\pi^i$ in $(s, a, c)$ according to Def.4.1. Intuitively, since contexts $c \sim \mu_C$ are fixed for each training episode, this reward encourages the algorithm to train policies that visit different state-action pairs within each context $c$, thus, promoting context-conditioned diversity.

### 4.1.1 Stabilising Diversity across Different Domains

Eq. 2 expands DOMiNO's repulsive force to operate across different contexts. However, the scale of intrinsic rewards varies greatly through different environments (i.e., different contexts will inherently yield different successor features). We need to stabilize fluctuations of such rewards when working with diverse contexts. We introduce two constraints to the learning objective that modulate the intrinsic reward $r_I$: a *dynamic intrinsic reward factor* $\chi$ that scales $r_d$ to target a factor of the moving average of the general extrinsic value and a *soft-lower bound* $\lambda$ that limits the search of diverse policies to a near-optimal subspace. We then define the DUPLEX intrinsic reward as:

$$r_I = \lambda \cdot \chi \cdot r_d \tag{3}$$

where $\chi$ and $\lambda$ are computed independently by the algorithm.

**Dynamic intrinsic reward factor.** $\chi$ scales $r_d$ proportionally to the sum of extrinsic values of policies in $\Pi$. Formally, $r_d \propto v_{e_{\text{avg}}} = \frac{1}{n} \sum_{i=1}^n v_{e_{\text{avg}}}^i$ with $v_{e_{\text{avg}}}^i = \alpha_{v_{\text{avg}}} v_{e_{\text{avg}}}^i + (1 - \alpha_{v_{\text{avg}}}) r_{e,t}^i$ where, $\alpha_{v_{\text{avg}}} \in [0, 1]$ weights the contribution of the extrinsic value of the i-*th* policy and its immediate extrinsic reward. Finally, $r_{e,t}^i$ is the extrinsic reward at $t$ when the agent acts under policy $i$. Intuitively, we are scaling the intrinsic rewards according to the average extrinsic value that the set of policies is achieving. Hence, at each algorithmic iteration, $\chi$ is updated as follows:

$$\chi_t = \alpha_\chi \chi_t' + (1 - \alpha_\chi) \chi_{(t-1)} \tag{4}$$

where $\alpha_\chi$ is the update rate of $\chi$, and $\chi' = |v_{e_{\text{avg}}} / v_{d_{\text{avg}}}|(1 - \rho)$ is the target value of the update, where $v_{d_{\text{avg}}}$ is the return value based on diversity intrinsic reward $r_d$ while $v_{e_{\text{avg}}}$ is based on the extrinsic reward $r_e$ instead. Through the optimality ratio $\rho$, $\chi$ minimizes the domain dependency by scaling the intrinsic rewards as a factor of the extrinsic objective and the optimality ratio $\rho$. Regarding the update rate $\alpha_\chi$, once we converged to a stable value, we kept it fixed across all experiments and domains.

**Soft-lower bound.** While $\chi$ preserves a relationship between extrinsic and intrinsic rewards, it does not prevent policies in $\Pi$ from exploring regions of the search space that are too far from the near-optimality regions of the target policies. To limit this possibility, we explored Lagrangian-constrained optimization but found it unsatisfactory in complex domains like GT. Hence, we introduce $\lambda$ to bound the near-optimal subspace for each policy using:

$$\lambda = \left\{ \sigma_k \left( \frac{v_{e_{\mathrm{avg}}}^i - \beta \hat{v}_{e_{\mathrm{avg}}}}{|\hat{v}_{e_{\mathrm{avg}}} + l|} \right) \right\}_{i=1}^n \tag{5}$$

where $\sigma_k(x) = 1/(1 + e^{-kx})$ and $\beta \in [0, 1]$ is a hyper-parameter indicating the reward region we are interested in exploring, $k$ regulates how "soft" is the bound, $n$ is the number of policies, $l$ is a small constant to prevent division by zero, and $v_{e_{\mathrm{avg}}}$ is the average extrinsic value. Intuitively, $\lambda$ limits the exploration of diverse behaviors to a near-optimal area defined by the threshold $\beta$. We find that introducing a sigmoid-based limit provides a more stable solution than Lagrangian-constrained optimization (See Figure 2). Similarly to $\alpha_\chi$, once we found a stable value for $k$, it remained fixed for all experiments.

## 4.2 Estimating Successor Features

Since we rely on $\psi$ to determine policy diversity (Def. 4.1), it is fundamental to estimate $\psi$ reliably. Additionally, as mentioned in Section 3, it is not tractable to keep $\psi^{\mathrm{avg}}$ for each context and task since they may be infinite. Thus building on top of UE, we exploit discounted expected features (SFs) $\psi^\gamma(s, a, z, c)$ and incorporate an extra head in the critic to estimate SFs $\tilde{\psi}^\gamma(s, a, z, c)$.

However, as in related work [30], we found that the critic struggles to correctly estimate $\psi^\gamma$ in complex settings. Specifically, since intrinsic rewards are bootstrapped over the expected difference of successor features, the accuracy in the SFs estimation plays a major role. Taking inspiration from the SAC algorithm [4], we incorporate an entropy term in the expected features objective to account for the stochastic component within the learned policies.

For policy $i$ in context $c$, our SFs at $s_t$ are computed as:

$$\psi^{\gamma,i}(s_t, a_t, c) = \phi_t + \mathbb{E}_{\pi_c} \sum_{k=t+1}^{\infty} \gamma^{k-t} \big[ \phi_k + \alpha_H H\big(\pi_c^i(\cdot|s, c)\big) \big] \tag{6}$$

where $\alpha_H$ is the entropy weight. Intuitively, unless the critic is confident that the actor will visit a state-action pair more frequently than others, the network will be encouraged to estimate that the policy will maximize the entropy. We formulate our temporal difference loss for the critic output $\tilde{\psi}^\gamma$ to account for the entropy term as:

$$\mathrm{TD}_{\mathrm{H}}(\tilde{\psi}_{\theta_j}^\gamma(\pi_c^z)) = \mathbb{E}_{(s,a)\sim\pi_c^z} \left[ \frac{1}{2} \left( \tilde{\psi}_{\theta_j}^\gamma(s, a, c, z) - y(\phi, s', c, z) \right)^2 \right] \tag{7}$$

where, similar to SAC, we have two critic estimates $j = \{1, 2\}$, but instead of taking the minimum of the two, our target is obtained from the average of the estimates. Then, we define $y(\phi, s', c, z)$ as:

$$y(\phi, s', c, z) = \phi(t) + \gamma \left( \operatorname*{avg}_{j=1,2} \tilde{\psi}_{\theta_{\mathrm{targ},j}}(s', \tilde{a}_z', c) - \alpha \log \pi_\omega^z(\tilde{a}_z'|s', c) \right) \tag{8}$$

with $\tilde{a}_z' \sim \pi_\omega^z(\cdot|s', c)$. The motivation lies in the different impacts of overestimation of values and successor features. When predicting values, overestimation directly influences the policy since it is trained to maximize advantage. However, in DUPLEX, SFs assume a distinct role: the policy is not incentivized to select actions with the highest successor features but rather those that differ the most from the SFs of the remaining policies (subject to not sacrificing too much performance, equation 1). Consequently, precision becomes of greater significance for $\psi^\gamma$, and taking the average emerges as a more reliable solution.

The introduction of entropy improves stability in the estimation of SFs through most of the training process and, as we report in the experimental section, it supports OOD generalization. However, as learning stabilizes, we typically encounter phases where the estimated difference between policies rapidly increases its order of magnitude. For this reason, we also introduce a fixed upper bound to $\alpha_\chi$ when estimating SFs. Details on the upper bound, DUPLEX pseudocode, and hyper-parameters are included in Appendix C and D.

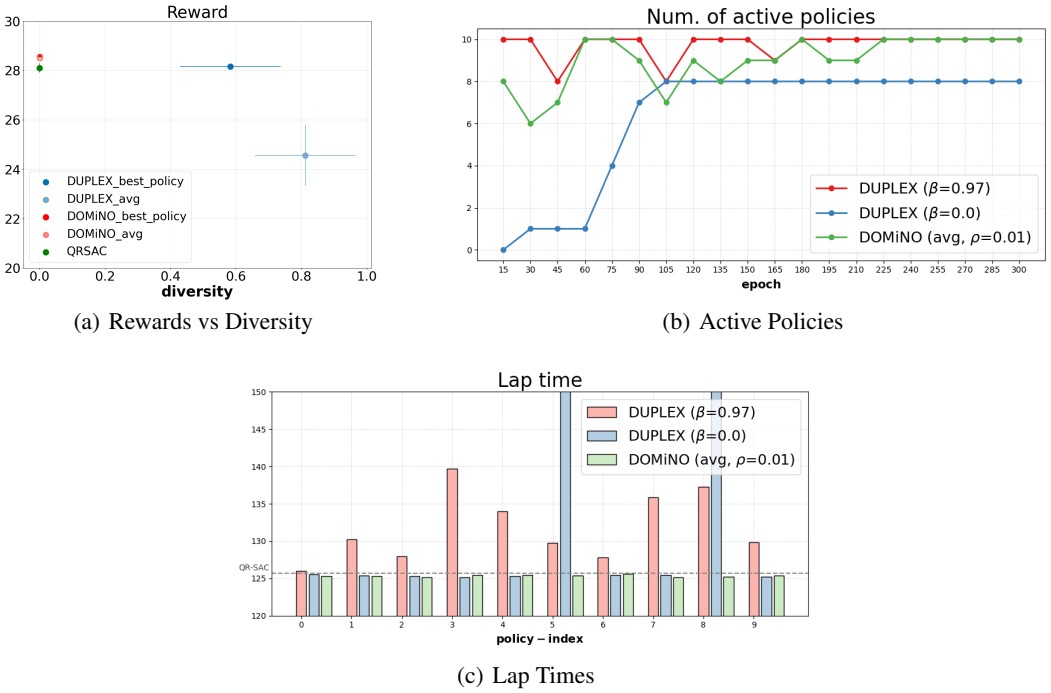

(a) Rewards vs Diversity

(b) Active Policies

(c) Lap Times

Figure 2: Results in GT. a) is the reward-diversity trade off, DUPLEX/DOMiNO_best_policy refers to $\max_{r_e}(\pi \in \Pi)$ of every run, while DUPLEX/DOMiNO_avg refers to $\text{avg}_{r_e}(\pi \in \Pi)$ of that run. Differently from DOMiNO, DUPLEX is able to generate *both* competitive and diverse behaviors. Figure b) illustrates the number of policies that finish laps (active). Figure c) min lap times of the diverse policies. Note that the number of active policies is important to showcase that all policies are actively searching for near-optimal diverse behaviors through different epochs. Applying our soft lower bound ($\beta \neq 0$) is key to focusing diversity in the region of interest.

Summarizing, DUPLEX enhances diversity learning in context-conditioned environments through four key components: (i) *dynamic intrinsic reward factor* ($\chi$) that balances diversity with extrinsic rewards and avoids environment-specific tuning; (ii) *soft lower bound* ($\lambda$) that constrains policies to optimize for diversity only within a target near-optimal region; (iii) *entropy regularization* ($\alpha_H$) in successor feature estimation to improve robustness; and (iv) averaging over critic estimates for computing diversity rewards. In Section 5 we demonstrate how these features improve both performance and diversity of the learned policies.

## 5   Experiments

The objectives of our experiments are to demonstrate that DUPLEX: 1) is the first successful method in learning diverse driving styles in highly-dynamic physics simulators such as GT (Figure 2); 2) when compared against previous state-of-the-art diversity approaches in RL, yields a better performance vs. diversity trade-off in canonical physics simulators (Figure 3); 3) exhibits diverse and effective behaviors in OOD dynamics and tasks (Figure 4); and finally, 4) improves on previous baselines in estimating SFs (Figure 5 and 6). All results in MuJoCO are obtained from five training runs. GT experiments are more computationally demanding requiring a minimum of seven days to converge, so the results are obtained from three training runs.

**Baselines.**   Our main baseline is DOMiNO [6], which is the state-of-the-art of diversity learning in RL. Then, since DOMiNO is not designed for OOD generalization, our experiments in OOD settings include as baselines the two most popular frameworks within UE literature: UVFA [8] and USFA [9]. Note that, throughout the section and given the two versions of DOMiNO, we will refer to

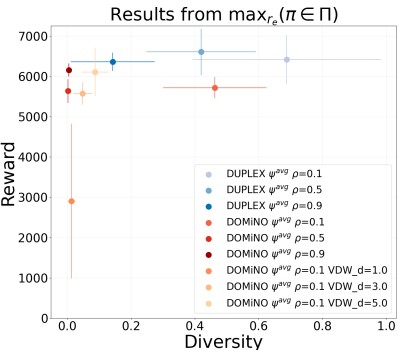
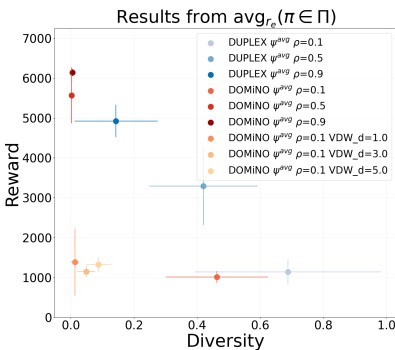

Figure 3: Walker single-task results. Both DUPLEX and DOMiNO do not estimate SFs and use $d_{\pi_c}^{avg}$. On the left, we show results from the best policy in each variant while on the right, the average reward across all policies. Dots and lines represent mean and standard deviation, respectively. It is worth reporting that the vanilla SAC implementation scores $5576.24 \pm 256.783$ when evaluated at the same training epoch.

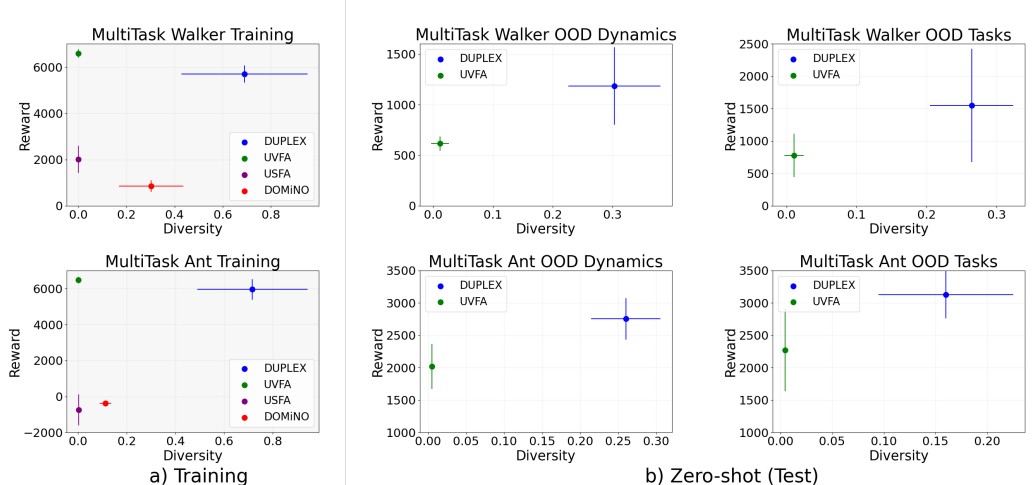

Figure 4: Results in MuJoCo Walker (Top) and Ant (bottom) multitask environments. On the y-axis, we report the reward accumulated by the agent while on the x-axis normalized diversity according to Def. 4.1. Ideally, we want to be as close as possible to the top right region. OOD dynamics refers to gravities that are at least 40% stronger or weaker than the strongest and weakest gravities seen in training, respectively. OOD tasks represent tasks where the agent needs to walk forward below a height 20% lower than the lowest height seen in training. Only DUPLEX and UVFA are evaluated in the Test since the other algorithms already failed in training.

DOMiNO$_{avg}$ as the version of the algorithm using averaged cumulants as successor features, and as DOMiNO$_{est}$ as the version of the algorithm that estimates SFs.

**Benchmarks.** We use three benchmarks. The first is a racing track in GT, where the goal is to have different driving styles while completing fast laps. Achieving diverse and competitive driving styles in such realistic simulator [12] can unlock multiple applications both in gaming and self-driving. Second, a set of experiments is conducted in the MuJoCo's Walker2D and Ant environments where we compare DUPLEX against DOMiNO$_{avg}$ in the same environments used in [6]. Finally, in the last benchmark, we evaluate our approach in a multi-context version of MuJoCo's Walker2D and Ant environments. We extend the default versions of these scenarios to include a three-dimensional context $c$ where the inputs represent: the gravity coefficient and, the upper and lower height thresholds we would like the agent to respect while moving. The goal of this benchmark is to find diverse policies in OOD contexts while guaranteeing competitive performance in task execution. Note that,

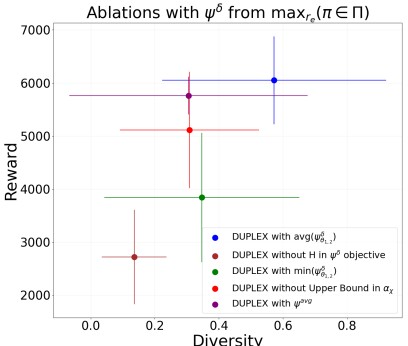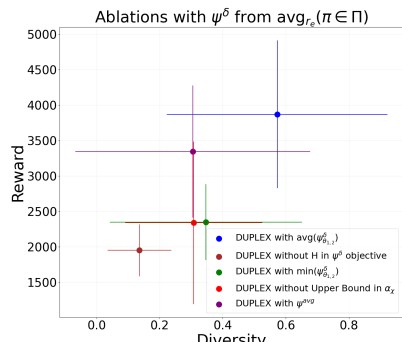

Figure 5: Ablation of DUPLEX in Multitask Walker. Reward and diversity scores of the best trained policy are reported on the left, while averaged values of all policies are on the right.

to have a fair comparison against the baselines, only the last benchmark configures DUPLEX to estimate SFs through $\psi^\gamma$. Additional details about the metrics and the environments are reported in Appendix A.

**Results in GranTurismo™ 7.** As in [12], we use QRSAC as the base RL algorithm for both DOMiNO$_{avg}$ and DUPLEX. Figure 2 (top-left) shows that only DUPLEX learns diverse policies in the near-optimal region. Figure 2(c) illustrates how this diversity translates into diverse lap times. Moreover, Figure 2(b) provides evidence of the benefits that our lower bound $\beta$ formulation carries with it. When using $\beta = 0$, the most diverse policies are "busy" finding different ways to *not* finish the track. Furthermore, we see DUPLEX results oppose to DOMiNO that instead collapses all policies to a similar behavior even when using an extremely forgiving value of the optimality ratio $\rho$. It is worth mentioning, that DOMiNO fails to provide diversity even if it implements a Lagrangian constraint optimization objective that should support more adaptability. For example, when configuring $\rho = 0.01$, DOMiNO should consider as "acceptable" any diverse policy achieving 1% of the value of the target policy $\hat{v}_{e_{avg}}$. But still, the algorithm struggles to find diverse policies.

**Results in MuJoCo.** To further validate the contribution of DUPLEX to diversity learning, we include a direct comparison of DUPLEX and DOMiNO$_{avg}$ while configuring the Walker2D environment that the authors present in [6]. Within this set of experiments, both algorithms have been configured to work with $\psi^{avg}$ which fosters the opportunity to ablate the benefits of our constrained $r_I$ (Eq. 3) isolated from the impact of estimating SFs. Figure 3 includes the results from both frameworks with multiple optimality ratios $\rho$. Moreover, to provide an exhaustive evaluation, we also configure DOMiNO$_{avg}$ with various Van Der Waal (VdW) distance hyper-parameters. Such a parameter modulates the desired distance between all the policies in $\Pi$ (see [6] for details). Finally, DUPLEX is able to find a better Pareto-frontier for the quality-diversity objective (Figure 3 right), while achieving better near-optimal policies than the baseline (Figure 3 left).

**Results in multi-context MuJoCo.** Figure 4 reports the results of our baselines in multi-context scenarios when evaluated within- and out-of training distribution. In the within-distribution setting (Figure 4 left-hand side) only DUPLEX shows good performance and diversity while USFA and DOMiNO fail at both learning competitive policies and providing diversity. As we report in the USFA ablation (B), DUPLEX succeeds in such benchmark due to the entropy term introduced to make the estimation of SFs more robust (Eq. 7). Additionally, note that USFA and UVFA do not optimize for diversity, and their diversity score is expected to be close to zero. Then, Figure 4 (right-hand side) reports the generalization effectiveness of DUPLEX and UVFA when operating out-of training distribution. Note that USFA and DOMiNO do not qualify in this setting as they failed to learn already in the training set. We observe that in all the OOD scenarios DUPLEX outperforms UVFA and, most importantly, it covers a bigger range of diversity values while completing the tasks.

**Ablation of successor-feature estimation.** Figure 5 illustrates the contribution of $\psi^\delta$ and the different features incorporated by DUPLEX. We have evidence that all features that compose DUPLEX are key to guaranteeing robust learning of diverse behaviors in highly-dynamic environments and

multi-context settings. Notably, USFA can also effectively solve multi-context MuJoCo environments when akin to DUPLEX, adds the entropy term (see Appendix B).

## 6    Conclusions

DUPLEX provides a powerful technique that enables learning of diverse near-optimal behaviors. Our experimental session shows that (differently from other baselines) DUPLEX (i) learns effective diverse behaviors in hyper-realistic complex domains (such as GranTurismo) and (ii) generalizes to OOD in challenging multi-context MuJoCo environments by demonstrating competitive diverse behaviors.

**Limitations and Future Directions.** There is potential to refine and enhance the methodology presented in this work. Nevertheless, we perceive the current limitations of DUPLEX not as setbacks, but as exciting opportunities that pave the way for new research avenues. For example, (1) can we generate more than twenty diverse policies and still guarantee meaningful diversity? (2) DUPLEX carries an additional cost due to the need to compute SF distances of each policy pair in our set, thus how can we improve sample efficiency and keep a low computational footprint? (3) DUPLEX does not impose any exploration strategy on each policy, and thus, can we control in what measure a particular policy will be different? And finally, (4) can we combine the strengths of different on a single solution? This latter question is very interesting to us. We suppose that once diverse policies have been learned, each of them retains different useful skills (e.g. different commuting routes) that can be combined dynamically to reconstruct a single agent capable of acting optimally and tackling unseen portions of the state-space.

Finally, the presented approach holds significant promise for advancing research in diverse policy learning, scalability, and efficient transfer and exploration [36, 37] across varied contexts. We note also that RL algorithms may also risk reinforcing biases and require careful implementation to ensure transparency and fairness. Diversity-learning algorithms might result in training policies with potentially unethical behaviors that are harder to predict. These are reasons that motivate us to pursue controllability while learning diverse policies in this and future works.

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

## A  Environment details

In MuJoCo environments, we measure diversity directly on the agent's observations, which is represented as a feature vector containing the angles and velocities of the agent's joints. In GT, diversity is measured through the most characterizing descriptors of a driving style: action intensities (brake, throttle, and steering values), wheel angles, and distances to the track edges.

Note that these diversity features were chosen to align with each environment's objectives. Nevertheless, we can maximize the diversity of any feature in the observation. Such an arbitrary selection of features intuitively leads to different optimization objectives and thus learned strategies. For instance, in GT, we could maximize diversity on the frequency of hitting other cars, and this would lead to a population of policies that are all driving as fast as possible while exploring the spectrum of aggressiveness: ranging from a very timid behavior (letting opponents pass to avoid contact) to a very aggressive one (hitting cars intentionally if the collision does not result in a significant speed loss).

We employ three types of environments to conduct our experiments. We use the canonical MuJoCo benchmark for single-task experiments; then the MountainTimeTrial track in GranTurismo™7 game for the GT experiments; and finally a MuJoCo wrapper environment for the multi-context experiments. Such a wrapper augments the observation of the agent by adding a context vector of three elements $c = \langle g, l, u \rangle$ representing the gravity coefficient $g$, the lower height the agent should walk below of $l$ and the upper height the agent should walk above of $u$. It is worth noticing that we excluded context configurations that would not be possible to solve, i.e. episodes were designed to let the agent learn to either walk low or jump high. At training time, in the Walker2D environment, the context features $\langle g, l, u \rangle$ take values in $g \in [6, 15]$, $l \in [0.8, 1.2]$ and $u \in [1.25, 1.8]$. While in the Ant environment features take value in $g \in [6, 15]$, $l \in [0.35, 0.8]$ and $u \in [0.9, 1.2]$.

In the OOD scenarios, we used two out-of-distribution contexts for the Walker2D and the Ant environments respectively. In the former, we used two values of the gravity coefficient and a lower bound constraint, namely $c_0 = \langle 3, 0.6, None \rangle$ and $c_1 = \langle 24, 0.6, None \rangle$, while in the latter we used $c_0 = \langle 3, 0.2, None \rangle$ and $c_1 = \langle 24, 0.2, None \rangle$

## B  Ablation in USFA

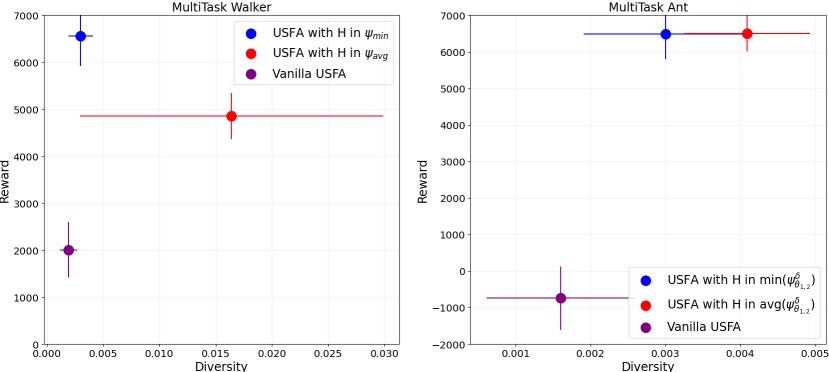

Figure 6: Ablation on USFA, we see that incorporating entropy to the learning objective of the successor features estimator as in DUPLEX yields a significant boost to accumulate rewards.

In this section, we ablate the addition of the entropy term to the SFs objective function. In particular, we extrapolate such component from the others introduced by DUPLEX, and we evaluate the entropy term in isolation within the original USFA framework. The assumption that we validate with this ablation study is that Equation 7 has a broader impact and that it makes the estimation of SFs more robust. Figure 6 illustrates our findings, we report that the entropy term enables effective learning in multi-context MuJoCo environments and significantly improves performance w.r.t. vanilla USFA.

Importantly, Figure 6 illustrates that, using $avg(\phi_{\theta_{1,2}})$ instead of the $min(\phi_{\theta_{1,2}})$, is not beneficial for USFA. Note that differently from DUPLEX, USFA infers extrinsic rewards from SFs. Thus, in USFA an overestimation of SFs (i.e., overestimating how often the policy visits some state-action

**Algorithm 1** DUPLEX

---

1: Initialize dataset with random exploration, set up critics $\theta$ and policy parameters $\omega$. and initialize moving averages $\{v^{\pi^i}_{\text{avg}}\}^n_{i=1} = 0$ reward weight $\kappa = 0$, $\chi = 1$ and $\text{step} = 0$

2: **while** not converged **do**

3:     Generate more data samples

4:     Sample batch of $m$ transitions $\{\text{t}_j\}^m_{j=1}$, where a transition is $\text{t}_j = (s^j, a^j, c^j, \phi(s^j, a^j, c^j), z^j, r^j_e, s'^j)$

5:     Use critic to get $\tilde{v}^{\pi^i}_{e_j}$, $\tilde{v}^{\pi^i}_{i_j}$ and $\tilde{\psi}^\gamma(s, a, c, z)$ for every $\pi^z \in \Pi$. Where $\tilde{\psi} = \underset{\text{h}=1,2}{\text{avg}} \tilde{\psi}_{\theta_\text{h}}$

6:     Compute $r_I(s)$ from $\tilde{\psi}, z, \phi(s, a, c)$ with Equation 3

7:     Compute extrinsic temporal difference (TD) errors and q-values: $\text{TD}_e, Q_e$

8:     For sampled $z$ get TD error with entropy on successor features with Equation 7: $\text{TD}_H$

9:     Compute intrinsic TD errors and q-values: $\text{TD}_I, Q_I$

10:     Combine q-values $Q = \sigma(\kappa)Q_e + (1 - \sigma(\kappa))Q_I$

11:     **if** $\text{step} > $ critic warming-up period **then**
         ▷ *Policy loss:*

12:         $\nabla_\omega \sum_j \left( \underset{\text{h}=1,2}{\min} Q_{\theta_i}(s, \pi_\omega(\cdot|s^j, c^j, z^j)) - \alpha \log \pi_\omega \left( \pi_\omega(\cdot|s^j, c^j, z^j) \big| s^j, c^j, z^j \right) \right)$

13:     Calculate weighted critic loss: $b_v(\text{TD}_e + \text{TD}_i) + b_\psi(\text{TD}_H)$

14:     **if** Constrained optimization **then**
         ▷ *Lagrange loss*

15:         $\sum^n_{i=1} \sigma(\kappa^i)(v^i_{\text{avg}} - \rho\hat{v}_{\text{avg}})$

16:         Update $\kappa$

17:     Update $\theta$ and $\omega$

18:     Update $v^i_{e_\text{avg}} = \alpha_{v_\text{avg}} v^i_{e_\text{avg}} + (1 - \alpha_{v_\text{avg}})r^i_{e,t}$, $v^i_{d_\text{avg}} = \alpha_{v_\text{avg}} v^i_{d_\text{avg}} + (1 - \alpha_{v_\text{avg}})r^i_{d,t}$, $\chi$ with Eq. 4 and $\text{step} \mathrel{+}= 1$

---

pairs) yields an overestimation of rewards. Such an overestimation motivates our design choice of using the $avg$ of two critics in DUPLEX. In fact, since DUPLEX relies on the distance between SFs to estimate intrinsic rewards, we achieve a more accurate representation of intrinsic rewards by taking the average of the two estimates.

## C   Training DUPLEX

DUPLEX follows an actor-critic approach, where a policy network determines actions to take while a critic network estimates values and, in the case of multiple contexts, also successor features. Figure 1 depicts the workflow of our algorithm. The networks have three independent input encoders for the state, policy, and context respectively. Since we use SAC as the base RL algorithm, we employ two critic networks and take the min of the two for value estimates $\tilde{v}^\gamma$ and the average for $\tilde{\psi}^\gamma$ (Eq. 8).

The training procedure of DUPLEX is described in Alg. 1. After warming up to collect data, we start sampling transitions $\text{t}$ and the learning routine (Line 4). Since we need to estimate the distance between all policies to compute the intrinsic rewards, we broadcast the input of the critic to have the SF estimates for all policies in every transition and do a forward pass through the critic (Line 5). Then, we obtain $r_I$ with Eq. 3 and compute the temporal-difference errors (Line 7 – Line 9). Similarly, after a short warm-up period, we compute the policy loss (Line 12). Then, depending on if we enable constrained optimization, we update the Lagrange multipliers (Line 15). Finally, we update the network weights and the average policy values to update the dynamic factor $\chi$ (Line 17 and Line 18).

**Upper bound on $\alpha_\chi$.** As described in Section.4, when the critic learns to estimate SFs, we typically observe events where the estimated difference between policies rapidly increases its order of magnitude. Such a shift in the magnitude of expected distances results in a rapid increase of intrinsic rewards, larger value estimates, and ultimately, the divergence of the critics output. To address such magnitude shifts, we modify the update rate of $\chi$:

$$\alpha_\chi = \begin{cases} 1 & \text{if } |v_{d_\text{avg}}| > \epsilon_\chi |v_{e_\text{avg}}| \\ \alpha_\text{default} & \text{otherwise} \end{cases} \tag{9}$$

Intuitively, we want $r_I$ to motivate the agent to explore increasingly diverse behaviors without stagnating the extrinsic rewards. For this reason, we need $\alpha_\chi$ to be a very low value – we use $1\text{e}{-}5$

across all our experiments. An exception is made when a sudden drastic change in $\tilde{\psi}$ could cause the critic to diverge. We automatically detect those changes through a threshold $\epsilon_\chi |r_e^{\mathrm{avg}}|$, where $\epsilon_\chi$ is a scalar, and perform a one-step update to prevent $r_I$ from destabilizing the learning process. Similarly to other hyper-parameters, once we found a stable value for $\epsilon_\chi$ it remained fixed for all experiments and benchmarks.

**Constrained optimization.** Within the algorithm framework, we allow Langrangian constraint optimization [38] to be contingent to specific domains. When constrained optimization is configured, we establish a multi-goal objective **(author?)** [39, 40] using Lagrange multipliers $\kappa$, whose objective is to maximize $\kappa(\rho \hat{v}_e - d_\pi \cdot r_e)$. Intuitively, this objective yields a multiplier $\kappa$ that decreases in value when the policy is satisfying the optimality constraint, and increases otherwise. However, we empirically find that while Lagrangian constrained optimization might be beneficial in canonical domains, it is not effective in complex environments such as GT and fails to find competitive diverse behaviors. Nevertheless, Equation (4) and 5 make our $r_i$ less reliant on constraint optimization than previous approaches. For these reasons, we implement constrained optimization in DUPLEX by keeping $\kappa$ fixed by default and making it learnable as an optional feature.

**Remark C.1** *DUPLEX is designed so that only two hyper-parameters significantly impact its performance: $\rho$ and $\beta$. The former regulates how much the user wants to weight the intrinsic rewards w.r.t. the extrinsic reward, while the latter defines the near-optimality region that the agent should explore when learning diverse policies.*

# D   Hyper-parameters and Computational Analysis

## D.1   Hyper-parameters

Table 1 lists DUPLEX hyper-parameters and neural network configuration.

Table 1: Hyper-parameters of DUPLEX.

| HYPER-PARAMETER | MUJOCO | GT |
|---|---|---|
| $N$ | 10 | 10 |
| $\rho$ | $[0.9, 0.7]$ | $[0.88, 0.8]$ |
| $\beta$ | 0.5 | 0.97 |
| $\alpha_\chi$ | $1e^{-5}$ | $1e^{-5}$ |
| $\epsilon_\chi$ | 20 | 20 |
| $k$ FROM SIGMOID | 5 | 5 |
| $\alpha_{\psi^{\mathrm{avg}}}$ | 0.9999 | 0.9999 |
| $\alpha_{v_{\mathrm{avg}}}$ | 0.999 | 0.999 |
| $\hat{v}$ SAMPLE RATE | 0.3 | 0.3 |
| $\kappa$ LEARNING RATE | $1e^{-2}$ | - |
| SAC/QRSAC LEARNING RATE | $3e^{-4}$ | $2.5e^{-4}$ |
| TARGET CRITIC TARGET SOFT UPDATE RATE | 0.005 | 0.005 |
| CLIP GRADIENT GLOBAL NORM | 3.0 | |
| OBS ENCODER SIZE | 256 | 2048 |
| CONTEXT ENCODER SIZE | 64 | - |
| POLICY ENCODER SIZE | 256 | 256 |
| TORSO SIZE | 2x256 | 4x2048 |
| CRITIC HEAD HIDDEN LAYER SIZE | 256 | 2048 |
| SAC ENTROPY LEARNING RATE | $1e^{-4}$ | FIXED ENTROPY |

## D.2   Computational Analysis

Given the additional components that DUPLEX requires to stabilize training, we report in the following table the computational cost of the update function of the core learning algorithm while varying the number of used policies. We compare our approach to DOMiNO and vanilla SAC and demonstrate that the additional cost is negligible when considering related work, and does not increase with the number of policies. Execution times of the selected algorithms are reported in milliseconds

and have been evaluated on the Walker2D MuJoCo environment rolled out on a 13th Gen Intel(R) Core(TM) i9-13900HX and NVIDIA GeForce RTX 4080 Laptop GPU.

Table 2: Execution time of the algorithms update function. Time is reported in milliseconds when computing diversity for 2, 4, 10, and 20 policies simultaneously.

| ALGORITHM | 1 | 2 | 5 | 10 | 20 |
|---|---|---|---|---|---|
| DUPLEX | | $55.78 \pm 1.51$ | $55.72 \pm 1.82$ | $55.21 \pm 1.37$ | $56.73 \pm 2.06$ |
| DOMiNO | | $52.64 \pm 1.88$ | $53.68 \pm 1.22$ | $54.64 \pm 1.36$ | $56.08 \pm 1.63$ |
| SAC | $41.01 \pm 2.05$ | | | | |

## E  Hardware

We ran our experiments on an internal cluster designed for distributed training. Each job has been configured with a trainer (to update the models) and N rollout workers (to collect data).

**MuJoCo.** In this environment we use one trainer and one rollout worker – none of them equipped with a GPU. The former uses 7.7 vCPUs and 8Gi of RAM, while the latter uses 1 vCPUs and 2Gi of RAM. The duration of each experiment is two days.

**GT.** In this environment, each job is configured with one trainer and ten rollout workers. The trainer is equipped with 7.7 vCPUs, 1 Nvidia V100, and 55Gi of RAM, while each rollout worker features 2000m vCPUs, 3328Mi of RAM, and 1 PlayStation4. The duration of each experiment is seven days.

