# OpenReview forum: "Discovering Creative Behaviors through DUPLEX: Diverse Universal Features for Policy Exploration"
_NeurIPS.cc/2024/Conference — NeurIPS 2024 poster_

### Official Review · Reviewer_XDei · 2024-06-24

**Soundness:** 3
**Presentation:** 3
**Contribution:** 3
**Rating:** 5
**Confidence:** 4

**Summary:**

This paper proposes an algorithm, DUPLEX, that learns diverse yet high-performance policies for context Markov decision process. DUPLEX extends the previous SOTA DOMiNO by introducing context in the MDP and extending the diversity objective as context conditioned. DUPLEX also introduces three tricks to stabilize the training: a dynamic scaling parameter to balance extrinsic and intrinsic reward, a lower bound to encourage high-performance policies, and a SAC-inspired entropy term to stabilize successor feature estimation. The experiment results show DUPLEX can learn diverse and high-performance policies on both highly complex GranTurismo environment as well as standard MuJoCo domains with OOD physics. In general, although the paper shows strong performance with its proposed method, I have concerns over the clarify of the paper and the evaluation.

**Strengths:**

1. The authors motivate the need for *diverse* high-performance policies well. I agree it is an important subject for research.
2. The paper presents several novel algorithmic ideas to stabilize the diversity-seeking policy training, including balancing the extrinsic and intrinsic rewards, a lower bound for encouraging being close to near-optimality regions, and a entropy term for stabling successor feature estimation.
3. The paper tests with the highly complex GranTurismo task and achieve impressive results. The authors also manage to show experiment results on MuJoCo standard tasks and ablation results, which help reproducibility.

**Weaknesses:**

1. The paper’s writing can be improved.

1.1. (Line 42-43) It seems to have a logical gap on why diversity in “multi-context setting” is key to enabling human-level task execution. As a result, I do not see why we should learn diverse behaviors on envs with different contexts (e.g., OOD environments).

1.2. (Line 132) the paper claims to find a set of diverse policies that maximizes the expected return in every context, but that does not seem to be enforced (not to mention there could be infinite c, which makes empirically showing that the policies are maximizing the return in *every context* impossible).

1.3. The Section 3 discussion about universal estimators seems disconnected with the intro and other texts around it. I do not understand from the description how universal estimators relate to the goal of finding diverse policies.

1.4. While I was able to follow Section 4, I find it hard. I suggest authors give a “preview” in the beginning of the section, and/or a diagram showing how different pieces of novel algorithms are connected together to form DUPLEX. It will also help motivate each algorithm component better.

1.5. The design of Equation 5 seems a mystery. I suggest authors provide stronger justification on the design, as well as exploring alternative, simpler design to achieve similar effect.

1.6. Similarly, the motivation for using an entropy term to stabilize successor feature estimation seems weak to me. Entropy term encourages the policy/value to respect uncertainties and multi-modalities in policy/value estimation, but SF estimation is unstable because of the Q-learning’s nature of chasing-its-own-tail learning and the difficulty to explore the entire space of hard problems such as GranTurismo. I do not see how they are related.

2. The features, as introduced in line 141 of the paper as “cumulants”, are essential to the core of the algorithm as involved in successor feature estimation and the diversity definition 4.1. However, the paper does not discuss how the features or cumulants are selected for any of the experiments, which makes me very concerned about reproducibility and how the algorithm works without very sophisticated feature choices. I strongly recommend the authors provide more details regarding it and if possible, ablation studies on different feature choices. Many of the experiment results could be trivial if DUPLEX is optimizing the defined diversity with the features, while baseline algorithms are optimizing different definitions of diversity and are evaluated with DUPLEX diversity.

Minor comments
1. Line 39 a period is missing.
2. While I agree successor features are good representation of a policy, I think it is also necessary to discuss the limitation of it – it ignores the order of features, which in some situations are essential. For example, successor features consider the state sequence a,b,c,a,b,c as the same as c,b,a,c,b,a, which is not necessarily always true.

**Questions:**

Please see the weaknesses for questions.

**Limitations:**

The authors did not discuss the potential negative societal impact of the work. I hope the authors consider including them in the rebuttal.

---

> ### Author Rebuttal · Authors · 2024-08-07
>
> We appreciate your feedback, it has been very useful to strengthen our work.
>
> **(Ln 42-43) It seems to have a..**
>
> Humans adapt OOD contexts continuously. For example, if a person injures their leg, they can immediately balance on one leg and even jump forward without using the injured leg. We aim to transfer this adaptability to agents in different contexts. For instance, a bipedal robot learning to walk should explore various walking modalities to be ready for unseen situations. This example is included in Line 43.
>
> **(Ln 132) the paper claims…**
>
> We agree that it is not possible to empirically prove that DUPLEX finds a set of near-optimal diverse policies for every context. However, we do not claim this. Instead, our training objective is designed to promote configurations that maximize the expected return across diverse policies in each context. Specifically, we assert that DUPLEX finds diverse policies and generalizes better to OOD contexts than previous state-of-the-art methods while maximizing the training objective (Line 132). We revised Line 131 to state: “The objective of our algorithm is to find a set of diverse policies that maximize the expected return in every context.”
>
> **The Section 3 discussion...**
>
> Our goal is to find diverse policies that can be generalized to different contexts. We make use of function approximators to handle cases where it would not be possible to enumerate all possible contexts that an agent will encounter. We replaced line 118 with:
> ```
> We briefly introduce the main building blocks of DUPLEX and the notation we use. First, we review basic concepts of multi-task reinforcement learning (RL) and explain how it can be represented by the contextual-MDP framework. Next, we describe the universal estimators used to enhance generalization across contexts, in cases where it would be hard to enumerate all possible contexts.
> ```
>
> **While I was able to follow Section 4…**
>
> We welcome the suggestion of the reviewer to improve clarity. We will lighten part of the preview at the beginning of Section 4.1.1 and introduce it at the beginning of Section 4 by describing how the different components are connected and interact.
>
>
> **The design of Equation 5 …**
>
> We explored various designs for Equation 5. Initially, we tried Lagrangian-constrained optimization but found it to be unstable, especially in domains like GT. Next, we tested a simple hard-threshold approach with `\beta \dot v_{target}`, but this made the reward signal discontinuous and more unstable. Ultimately, we used a sigmoid function to modulate the lambda parameter, as shown in Equation 5. This approach provided a continuous reward signal and effectively penalized deviations from near-optimality. Normalizing by `|v_{target} + l|` helped us avoid fine-tuning the sigmoid slope and maintain consistent hyperparameters across domains. We included this description in Appendix C.
>
>
> **The motivation for using an entropy…**
>
> Instability in RL often arises from predicting a moving target. Additionally, the complexity of training can increase depending on the application and task. We found that standard successor feature (SF) prediction is inadequate in multi-context settings.
>
> We hypothesized that adding an entropy term to SF estimation helps the critic to account for domain uncertainty, similar to the motivation behind entropy in standard RL algorithms like SAC. This approach stabilizes estimation and prevents divergence, as shown in Figure 5. We further explore this in Appendix B and Figure 6, where we demonstrate that adding an entropy term also improves the standard USFA framework, achieving a 3x improvement in multi-context MuJoCo domains.
>
> **The features…**
>
> Definition 4.1 offers a general definition of policy diversity, aligning with related work [2] and it does not introduce a novel diversity metric (we clarified this in the paper). That said, to evaluate DUPLEX against other baselines, we avoided feature engineering and, in all MuJoCo experiments, used the entire observation vector for cumulants without selecting specific features.
>
> In GT,  we selected features that better characterize a driving style: action intensities (brake, throttle, and steering values), wheel angles, and distances to the track edges and cars. It is important to mention that in this type of domain, the observation vector includes hundreds of features and it might be difficult to tease out behaviors that can be visually appreciated to the naked eye. Initially, we did use the whole observation also in GT, and DUPLEX was indeed able to find much better diversity trade-offs, but, it was finding diversity also over less relevant features such as how much dirt there is on the tires, or how much tires are on tarmac and how many on the grass.
>
> **Minor comments…**
>
> We appreciate the reviewer's comments. While we acknowledge the limitations of SFs, we did not find them to hinder our ability to discover diverse behaviors. This is supported by the fact that our approach remains effective even when using averaged cumulant values along a trajectory. That said, it is an interesting future direction that could lead to improving the accuracy of our agent's predictions. As a first step, we could explore n-step returns when computing the prediction errors. We included this observation in the conclusions when discussing limitations.
>
> **Negative societal impact:**
>
> We will add: RL algorithms optimize decision-making in various domains. However, they also risk reinforcing biases and require careful implementation to ensure transparency and fairness. Diversity-learning algorithms might result in training policies with potentially unethical behaviors that are harder to predict. These are the reasons that motivate us to pursue controllability while learning diverse policies (Section 6, Question 4).
>
> We thank the reviewer for their feedback, let us know if you have any further questions.
>
> [1] Haarnoja, Tuomas et al. 2017.
> [2] Zahavy, Tom, et al. 2023.

---

> > ### Comment · Reviewer_XDei · 2024-08-11
> >
> > Thank you to the authors for the detailed answers to my concerns and questions. I think the paper's clarity has been improved and my major concerns have been answered/mitigated, especially weakness2 (about the features), as the authors actually did not do any feature engineering for the simpler domains. As such, I increase my score to 5: borderline accept.
> >
> > I am still not quite buying the argument for entropy: SAC introduces entropy to maximize, i.e., finding policies that achieve high task reward and high diversity (entropy). However, what does it mean to maximize entropy in SF estimation? The key point is that you are trying to calculate/estimate SF, not to maximize them. It is very interesting to see introducing entropy also helps baselines, and I hope to hear more insights from the authors and also in the paper.

---

> > > ### Author Response · Authors · 2024-08-12
> > > **Follow up**
> > >
> > > Thank you to the reviewer for their response and increasing their score.  Regarding the addition of the entropy term:
> > >
> > > Indeed incorporating entropy to the critic estimators is not the same as incentivising the actor to maximize entropy. Our statement citing SAC was meant to credit our source of inspiration when ideating what would be done to master the multi-task challenge (as seen in Figures 4 and 5 and Appendix B, without this only UVFA, that does estimate SFs was able to learn competitive policies in this environment).
> > >
> > > Our intuition is that by maximizing the entropy in the estimation of what the actor will do, i.e. max entropy in SFs estimation, minimizes the critics’ underlying biases. That is, unless the critic is confident that the actor will visit a state-action pair more frequently than others, the network will be encouraged to estimate that the policy will maximize the entropy.
> > >
> > > The question that arises is the generality of the observation that the entropy maximization by default is beneficial. This is an interesting question since the underlying policies are also maximizing entropy, i.e., giving equal probabilities to all the actions when a dominant one is not available. Future research could further investigate this direction evaluating the performance of  USFA with other RL policies whose actor is not maximizing the entropy, e.g., DQNs.

---

> > > > ### Comment · Reviewer_XDei · 2024-08-13
> > > >
> > > > Thank you to the authors for answering my question about the entropy term. I think "unless the critic is confident that the actor will visit a state-action pair more frequently than others, the network will be encouraged to estimate that the policy will maximize the entropy" is a great intuition to explain the motivation of introducing entropy in SF. I suggest making that clear in future paper revisions.

---

### Official Review · Reviewer_VFg8 · 2024-07-08

**Soundness:** 3
**Presentation:** 2
**Contribution:** 3
**Rating:** 6
**Confidence:** 3

**Summary:**

This is a work in the Reinforcement Learning domain, particularly relevant to policy exploration. They propose a method to better preserve the trade-off between the exploration diversity and near-optimality. This strategy utilizes a diversity objective with defined constraints such that it enforces the trained agent learns a diverse set of near optimal policies and could exhibit competitive and diverse skills in out-of-distribution environment. Two experimental domains are used to showcase the superiority of the proposed method.

**Strengths:**

* The problem the author studies is an interesting and challenging direction. There are many related work and the author has clarified the motivation and their own contribution pretty clearly.
* The proposed method is technically novel and sound.
* The experiments performed are comprehensive.

**Weaknesses:**

* Relying on multiple hyper-parameters (introduced by equations 3-5, probably hard to scale across the domain).
* Some of the paragraphs has heavy notations and might be more readable if the author could give more intuitions on why it came out like that. For example in section 4.1.1 and section 4.2.

**Questions:**

* Regarding the diversity definition in Def 4.1, can you give an example of whether/how different modalities of observed features (e.g., visual or textual) could have a different impact on the diversity formula? or it does not matter (no effect).

* What is the relative additional cost introduced by computing SF distance of each policy pair?

**Limitations:**

I like how the author clarified their limitations in section 6. I am very interested in learning more explanations on question #3 and #4.

---

> ### Author Rebuttal · Authors · 2024-08-07
>
> We thank the reviewer for their interest in our work and the feedback provided. Below are the detailed responses:
>
> **Relying on multiple hyper-parameters (introduced by equations 3-5, probably hard to scale across the domain).**
>
> As indicated in lines 207-209 and 843-845, most of DUPLEX hyper-parameters do scale across domains used in our evaluations. As reported in Table 1, apart from network size and learning rate, to guarantee diversity, we only need to adjust `\rho` and `\beta`. It is worth mentioning that we want to expose such hyper-parameters to the user to be able to specify the performance vs. diversity trade-off according to the use case.
>
> **Some of the paragraphs has heavy notations and might be more readable if the author could give more intuitions on why it came out like that. For example in section 4.1.1 and section 4.2.**
>
> Reviewer XDei expressed a similar comment and suggested adding a preview paragraph describing how the components interact with each other at the beginning of Section 4. We, therefore, redirect this comment to that answer as well.
>
> **Regarding the diversity definition in Def 4.1, can you give an example of whether/how different modalities of observed features (e.g., visual or textual) could have a different impact on the diversity formula? or it does not matter (no effect).**
>
> This is an interesting observation. We believe that different modalities do not alter Definition 4.1, which holds valid. We can always compute L2-norm distances of tensors.
>
> However, under different modalities, it is important to understand how to design cumulants and guarantee that the agents focus on important features. Similar images can represent different things and the same concept can be expressed with very different words. For example, in GT, few pixels can draw the line between a mispredicted collision and an actual collision between cars.
>
> One approach is to preprocess visual and textual information with an encoder, or we could design modality-invariant cumulants. Nevertheless, guaranteeing execution under different modalities is an interesting aspect that is mandatory to generalize DUPLEX to diverse domains, and will be a topic in our future work.
>
> **What is the relative additional cost introduced by computing SF distance of each policy pair?**
> We will compute the execution time and report the additional cost of estimating SFs per batch iteration. It will be included in Section D. The computation of the distance of all the samples in the batch is done through matrix multiplication, not iteratively.
>
> **I like how the author clarified their limitations in section 6. I am very interested in learning more explanations on questions #3 and #4.**
>
> We appreciate your comment and your interest in our future work. Below we detail questions 3 and 4:
>
> Q.3) DUPLEX does not impose any exploration strategy on each policy …
>
> By design, DUPLEX does not bias exploration within the near-optimal regions and policies can converge to any local-minima in such space as long as the diversity constraint is satisfied. As a result, two consecutive experiments within the same domain may return with a different set of diverse policies. However, for some use-cases it is reasonable to guarantee that some behavior is always discovered. For instance, in Walker2D, there is always a policy that learns to jump by only using the left leg. In GT, an aggressive policy is often learned. In future work, we could explore how to differently mask cumulants for each policy in the set and bias exploration. Alternatively, we could design SF vectors that represent desired behaviors and use such vectors as behavior baselines for each policy.
>
> Q.4) can we combine the strengths of different on a single solution? …
>
> Intuitively, even though the target policy is configured to only maximize the extrinsic rewards, the other policies (due to the added exploration) could converge to behaviors that are more efficient in specific state transitions. We believe that enhancing the target policy with situational corrections coming from the auxiliary policies can improve the overall performance of the agent.
>
> In early experiments, we explored generalized policy improvement (GPI) [2], which intuitively, uses the critic estimations to select, at each step, the best action over a set of policies. However, empirically, we did not find it to improve the performance of our agent. We hypothesize that the imperfect information of the critic becomes especially relevant in continuous-action environments. Moreover, our continuous settings might demand GPI to be executed with a longer horizon, and a “simple” step-wise composition of diverse policies might not be sufficient.
>
> Finally, we believe that improving accuracy in the critic estimations and taking advantage of the added exploration of diverse learning approaches is an exciting line of work and is necessary to advance the research field.
>
> [1] Wurman, Peter R., et al. "Outracing champion Gran Turismo drivers with deep reinforcement learning." Nature 602.7896 (2022): 223-228.
> [2] Barreto, Andre, et al. "Transfer in deep reinforcement learning using successor features and generalized policy improvement." International Conference on Machine Learning. PMLR, 2018.

---

### Official Review · Reviewer_9gQo · 2024-07-16

**Soundness:** 2
**Presentation:** 3
**Contribution:** 1
**Rating:** 5
**Confidence:** 4

**Summary:**

This paper proposes to use successor feature (an embedding of state-action pair) to boost the behavior diversity of a population of policies.

Several tricks are proposed to compute the diversity intrinsic reward. These include using the running average of the extrinsic reward to scale the intrinsic reward and using a dynamic weight $\lambda$ to ensure the task performance is not dropped too much.

Experiments on GranTurismo 7 and Mujoco show that the method can learn diverse yet high-performing policies, which also shows certain OOD capacity.

**Strengths:**

The proposed method is simple.

The detailed ablations on each part of the proposed method is sufficient.

The behavior in GranTurismo 7 is interesting and it's indeed diverse yet competent.

**Weaknesses:**

There is a strong assumption of the method: the context vector is given by the environment. The claim on context-conditioned MDP doesn't seem to be interesting to me as considering those context vector as part of the observation space will not change anything. Thereafter the claim in Line 174-175: "We adopt the repulsive reward ... and extend it to be conditioned on the context" does not mean a lot and it suggests the technical novelty of the method against Zahavy is limited.

The method is not scalable as you need to compute Equation 2 and Equation 3 for every state action pair to collect the intrinsic reward. That means you need to iterate over all agents to get their successor feature.

No code is provided.

**Questions:**

What does "closest expected features" in Line 178 mean? A feature is a vector, how you get the closest vector of another vector? Are you computing the L1/L2 norm as the distance between features?

Clarity issues:
* Typo in Line 198 "or the update"
* There should be a $i$ notation in $r_d(s, a, c)$ in Eq 2 to indicate that reward is for policy $i$.
* Equation 7 has an undefined $y$.

What does the braces in Eq 5 means?

How $\phi(s, a, c)$ is computed? The output of a neural network? In Appendix Algorithm Line 5 you say "use critic to get" it, what does this mean? Please describe the detailed architecture of the critic network.

It seems the method use QRSAC as the RL algorithm. Will you recompute the intrinsic reward for each sampled batch during training? Or you store the intrinsic reward as it is to the replay buffer?

Missing citations to relevant works:

* Exploration by Random Network Distillation
* Non-local policy optimization via diversity-regularized collaborative exploration

**Limitations:**

Limitations are addressed.

---

> ### Author Rebuttal · Authors · 2024-08-07
>
> We thank the reviewer for the time and appreciate the feedback provided.
>
> **There is a strong assumption of the method: the context vector is given by the environment.**
>
> We agree that not all environments are designed to provide a context vector. However, we comply with related work in zero-shot generalization [1] where contextual information is provided as a separate input that is procedurally generated by the environment. That said, we believe that extending our work to domains with hidden contextual information is a promising future direction and will include it in the conclusion section. However, it is out of the scope of this paper.
>
> [1] Kirk, Robert, et al. "A survey of zero-shot generalization in deep reinforcement learning." Journal of Artificial Intelligence Research 76 (2023): 201-264.
>
> **The claim on context-conditioned MDP…**
>
> We agree that only extending Equation 2 to contextual-MDPs is not a significant contribution and thus it is not listed as a main contribution. To have a fair comparison, we also provide contextual information to DOMiNO when evaluating it.
>
> To highlight the novelty, we list the technical contributions briefly here. These are detailed in Section 4 and the ablation studies.
> Dynamic intrinsic reward factor to modulate the ratio between diversity and extrinsic rewards, alleviating the reward tuning that DOMiNO requires across different environments
> Soft-lower bound to limit the region of interest for the diverse policies.
> Entropy regularisation term in SFs estimation to improve the estimation of SFs in multi-task environments.
> Using the average of the critic estimates is beneficial when estimating SFs and at the same time computing diversity
>
> To further highlight the novel components that DUPLEX introduces, we added a brief paragraph summarizing key contributions at the end of Section 4. As evidenced in the ablations, all these components have a significant impact with respect to DOMiNO and enable our agent to achieve competitive performance in hyper-realistic domains and in multi-task environments.
>
> **The method is not scalable…**
>
> Please note that Equation 2 and Equation 3 are not computed separately for each pair of policies but instead are vectorized through matrix multiplication, and thus scalable. We note that the equations are computationally equivalent to the equations in Zahavy’s method (which provides pseudocode on how to compute these equations as a matrix). We specified in Appendix C that we do not increase computational complexity with respect to previous baselines.
>
> **No code is provided.**
>
> We cannot share the code to this date. We included a detailed pseudo-code description of our algorithm, described our environment domains, and included the hyper-parameters used for training DUPLEX agents.  We believe this to be the key information needed to reproduce our experiments but we are happy to include further details otherwise.
>
> **What does "closest expected features" …**
>
> According to Definition 4.1, we compute distances among policies as the L2-norm distance between their feature vectors. We added a reference to Definition 4.1 on line 178.
>
> **Clarity issues**
>
> We thank the reviewer for the detailed review. We have integrated the recommended corrections into the equations. Note that the definition of `y` follows in Equation 8 and we have updated the text to better describe this connection.
>
> **What does the braces in Eq 5 means?**
>
> Braces indicate that the soft-lower bound is defined as a vector of n = size(\Pi) components, one for each policy. We corrected a typo in the equation that as currently defined, would consider n + 1 policies.
>
> **How  ϕ(s,a,c) is computed? The …**
>
> We compute expected features ψ by predicting the expectation of the cumulants ϕ through the estimation of the critic network. And we achieve that by adding an extra head to the critic network.
>
> More specifically, cumulants ϕ are a part of the input vector where we want to maximize diversity. By default, we use the whole environment observation as ϕ. Nevertheless, if we want to tease out diversity in particular features, as in the case of GT, we only use of subset of the observations to define ϕ. For example, in the aggressiveness scenario shown here https://sites.google.com/view/duplex-submission-2436/home, only the part of the observation that characterizes collisions among cars is included in ϕ, and as a result, the agent trains policies with different levels of aggressiveness.
>
> Expected features ψ are directly estimated by an additional head of the critic network (Line 8) and the error is obtained through Equation 7. The critic architecture is detailed in Figure 1. and the size of the expected features vector is equal to the number of cumulants. We included this description in Table 1.
>
> **It seems the method use QRSAC…**
>
> We use SAC in the canonical MuJoCo environment for repeatability of the results and use QRSAC in GT as it represents the state-of-the-art in that domain.
>
> Since the intrinsic reward depends on the transitions sampled from the environment, we compute the reward signal for each batch. Nevertheless, this operation is vectorized through matrix multiplication and it is not iteratively computed, which would make the approach impractical.
>
> **Missing citations to relevant works**
>
> We are happy to include these references, we also think that exploration is key when learning diversity and these citations represent a promising future direction to further improve our method.

---

> > ### Comment · Reviewer_9gQo · 2024-08-09
> >
> > Thanks for the response. I changed my score to 5.
> >
> > We all argree that the "contextual MDP" and "context-conditioned diversity learning" is not a strong claim and novelty of the paper. My major concern is the technical novelty of the paper as well as the research problem. The paper seems like a combination of existing works, some heurisitics (e.g. soft-lower bound), on a less-interested tasks: the diversity in mujoco, especially in 2024 is it really an important problem?
> >
> > The paper itself though is complete and sound.

---

> > > ### Author Response · Authors · 2024-08-12
> > > **Follow up**
> > >
> > > Thank you for your response and updating your score. Regarding your concerns on the technical novelty and significance of our contribution, please note:
> > >
> > > - Research significance:
> > >   - Our research is the first one to learn end-to-end competitive diverse policies in hyper realistic racing simulators and to show visibly diverse driving styles. As reported in the paper, previous state-of-art for diversity in RL failed to provide any visually perceptive degree of diversity while guaranteeing competitive behaviors.
> > >
> > >    - Demonstrating near-optimal performance while combining diversity and multitask generalization has direct applications to the GT player community, increasing replayability and engagement for users by racing against diverse and competitive opponents. For instance, we invite the reviewer to check https://sites.google.com/view/duplex-submission-2436/home#h.57pf96h6i4l7 in the first GranTurismo video in the link, and go to the timestamp 1:34 to contrast policies 0 and 1 in their overtake maneuver. It can be seen how the different policies that were so far driving quite similarly take very different strategies to overtake. Having opponents that use such diverse complex tactics can be very rewarding for players and has practical applications for self-play in RL.  Moreover, being robust towards OOD tasks and environments makes these agents more applicable to new game functionalities and updates.
> > >
> > >    - We are first to introduce an RL algorithm that provides multiple and diverse solutions in OOD tasks and environment dynamics. As demonstrated in Figure 4, where DUPLEX generalizes better than UVFA and USFA, diversity can be helpful in finding better solutions to OOD tasks and dynamics.
> > >
> > > - Technical novelty:
> > >    - Our algorithm builds upon previous state-of-the-art methods from two disconnected bodies of research in RL, several simple yet effective heuristics, and a novel objective function for one of the main components. Simplicity in our method should not hinder the merit of our work, since we prove how our algorithm outperforms related work, and how all of the DUPLEX components play a fundamental role that compounds to achieve the final result (see ablations in Figures 2 and 5). Moreover in Appendix B, we demonstrate how our objective function to estimate successor features is beneficial to previous state-of-the-art frameworks such as USFA.
> > >     - Finally, the multi-task Mujoco proved to be a valid and challenging benchmark for current state-of-the-art methods such as DOMiNO and USFA, that failed already in the training set.

---

### Official Review · Reviewer_Eqtv · 2024-07-19

**Soundness:** 2
**Presentation:** 3
**Contribution:** 2
**Rating:** 5
**Confidence:** 3

**Summary:**

The authors introduce DUPLEX, a method that defines a diversity objective with constraints and uses successor features to robustly estimate policies' expected behavior. DUPLEX allows agents to:
1.	Learn a diverse set of near-optimal policies in complex, highly-dynamic environments.
2.	Exhibit competitive and diverse skills in out-of-distribution (OOD) contexts.
The authors claim that DUPLEX is the first method to achieve diverse solutions in both complex driving simulators and OOD robotic contexts.

**Strengths:**

a. The diversity objective is commendable as it promotes the development of a set of policies that are diverse enough without hindering reward maximization.

b. The paper is very well-written and organized. The logical structure of motivation, related work, background, and analytic study of the frameworks, combined with the metric and the presentation of the algorithm and its evaluation, is easy to follow and intuitively explained. The inclusion of small examples to illustrate the arguments helps lighten otherwise densely formulated reasoning.

**Weaknesses:**

•	The paper does not compare its diversity objective to other existing diversity objectives in the literature, such as DIAYN (Eysenbach et al., 2019) and MaxEnt RL (Haarnoja et al., 2018). This omission makes it difficult to judge why this particular diversity measure should be considered over the others.

•	The paper builds upon the Domino paper, but it is not clear how it differs from Domino since both papers use extrinsic rewards and another metric computed to promote diversity, i.e., intrinsic reward.

•	Another fundamental question arises: If a model can generate stochastic policies with high rewards, will it be considered diverse compared to the deterministic policy model? Is diversity essentially a measure of generating high-reward stochastic policies?

•	I would urge authors to upload code and some visualizations with comparison of the method to Domino during rebuttal period.


Overall a good paper and I would be happy to discuss more with authors. I will be willing increase my score if my questions are answered.

**Questions:**

See Weaknesses

**Limitations:**

Authors have mentioned the limitations in the paper.

---

> ### Author Rebuttal · Authors · 2024-08-07
>
> We appreciate the time and feedback you provided on our submission.
>
> **The paper does not compare its diversity objective to other existing diversity objectives in the literature, such as DIAYN (Eysenbach et al., 2019) and MaxEnt RL (Haarnoja et al., 2018). This omission makes it difficult to judge why this particular diversity measure should be considered over the others.**
>
> Including algorithmic baselines can greatly enhance a scientific contribution. For our experimental section, we selected baselines that we believe best support the claims of our work. We consider DOMiNO to be the state-of-the-art method in reward-based diversity learning and thus encompasses other baselines.  In contrast to DIAYN and MaxEnt RL (and DOMiNO), DUPLEX is also being evaluated in OOD multi-task generalization. In such a setting it would not be a valid data point for the comparison against standard diversity learning algorithms. To support the contribution of DUPLEX in such settings, we rely on state-of-the-art frameworks for multi-task learning such as UVFA and USFA.
>
> **The paper builds upon the Domino paper, but it is not clear how it differs from Domino since both papers use extrinsic rewards and another metric computed to promote diversity, i.e., intrinsic reward.**
>
> As the reviewer correctly points out, DUPLEX’s core definition of diversity and diversity reward computation is aligned with DOMiNO. However, we extended the training problem to include context-based scenarios and enhanced the robustness of training diverse policies. We achieved this by adding new components, which are briefly outlined here (and detailed Section 4):
> - *Dynamic intrinsic reward factor* to modulate the ratio between diversity and extrinsic rewards, alleviating the reward tuning that DOMiNO requires across different environments
> - *Soft-lower bound* to limit the region of interest for the diverse policies.
> - *Entropy regularization term in SFs estimation* to improve the estimation of SFs in multi-task environments.
> - *Use of the average of the critic estimates*. Differently than taking the min of the critics as it is common with value estimation, we find that to compute diversity rewards is beneficial to take the average of the estimated SFs.
>
> To further highlight the novel components that DUPLEX introduces, we added a brief paragraph summarizing key contributions at the end of Section 4.
>
> **Another fundamental question arises: If a model can generate stochastic policies with high rewards, will it be considered diverse compared to the deterministic policy model? Is diversity essentially a measure of generating high-reward stochastic policies?**
>
> In this work, according to Definition 4.1. we consider (deterministic and stochastic) policies to be “diverse” if they feature diverse state-action occupancies ). A stochastic policy will visit different state-action pairs than a deterministic one. Hence these two policies can be considered diverse. When introducing a near-optimal constraint, we are still searching for policies that feature diverse state-action occupancies, but we constrain the search within a particular area of the search-space, which is defined by Equation 1.
>
> In other words, the reward obtained by the agent (or model) has no impact on the computation of the diversity metric, but since DUPLEX enforces that all the policies have to simultaneously optimize for the true objective as well, it yields the ability to maximize discounted cumulative rewards (high-rewards).
>
> **I would urge authors to upload code and some visualizations with comparison of the method to Domino during rebuttal period.**
>
> We cannot share the code to this date. We included a detailed pseudo-code description of our algorithm, described our environment domains, and included the hyper-parameters used for training DUPLEX agents.  We believe this to be the key information needed to reproduce our experiments but we are happy to include further details otherwise.
>
> We are working towards including visualizations of Domino vs DUPLEX in our site https://sites.google.com/view/duplex-submission-2436/home but, we will likely add them after the discussion period ends.
>
> In the meantime, we can describe our observations from completed experiments. DOMiNO typically produces policies with subtle differences, as seen in Figure 1(b) of [2], where each policy raises the leg slightly more than the previous one. Occasionally, by sufficiently reducing the optimality ratio, we may observe two “main behaviors” that are visually distinguishable, with other policies showing small variations around them. In contrast, DUPLEX tends to produce significantly more diverse populations of near-optimal policies that are visually distinct. This greater diversity yields better trade-offs, as shown in Figures 2, 3, and 4.
>
> Thank you for your useful feedback. We will be happy to respond to any further questions you might have.
>
> [1] Zahavy, Tom, et al. "Discovering Policies with DOMiNO: Diversity Optimization Maintaining Near Optimality." The Eleventh International Conference on Learning Representations.

---

### Official Review · Reviewer_zFsC · 2024-07-29

**Soundness:** 2
**Presentation:** 2
**Contribution:** 2
**Rating:** 5
**Confidence:** 3

**Summary:**

introduced a method to enhance diversity in RL policies by using successor features for robust behavior estimation. Experiments show DUPLEX outperforms state-of-the-art baselines, achieving diverse, competitive policies in GranTurismoTM 7 and multi-task MuJoCo environments, even in out-of-distribution contexts.

**Strengths:**

1. The ability to learn multiple strategies for a given task is meaningful and can shed light on future decision-making agents' research.

2. The experimental results show that the proposed method outperforms previous works in complex driving simulators and OOD robotic tasks.

**Weaknesses:**

I am not familiar with work related to universal function approximators, but I am knowledgeable about quality diversity and RL-related work. Therefore, I will focus on raising questions about the latter.

1. In 4.1.1, the authors mention, "We explored Lagrangian-constrained optimization but found it unsatisfactory in complex domains like GT." Although this statement is experimentally validated, many past QD RL works have used Lagrangian-constrained optimization successfully[1,2]
, making this claim counterintuitive to me. Could the authors compare their results with these works or explain in detail the reasons behind their findings? Furthermore, could they mathematically explain the impact of multiplying the reward function by lambda on the optimization objective?

2. In the experimental section, the right side of Figure 3 shows that to improve the diversity score by 0.15, the reward decreases to 5000 points, which is not very high for the walker task. I have not run diversity strategy experiments under this setting and thus lack a reference point. However, I feel the increase in the diversity score is insufficient. Could the authors run n instances of SAC/PPO, each with different seeds, and stop at around 5000 points to report the diversity scores as a reference? Otherwise, the Baseline (DOMiNO) scores in the lower left of Figure 3 make me suspect the authors have not correctly implemented the baseline algorithm. The same issue appears in Figure 4, where DOMiNO is in the lower left, and I would like a reasonable explanation from the authors.

3. There are several minor errors in the paper, which, while partially understandable in context, collectively make the paper difficult to understand:
- $V_{d_{avg}}$ is not defined (line 196).
- $V_{e_{avg}}$ is first mentioned in line 191 but introduced only in line 207.
- The SF estimator on the left side of Equation 7 includes $\gamma$, but the right side does not.
- $y$ in Equation 8 is not a function of $c$, yet $c$ is used on the right side.
- The $z$ in Equation 8’s right side is in the superscript, where $\gamma$ was used earlier.
- The legend in Figures 3 and 5 is too small.
- The difference between the left and right of Figure 5 is not well-explained.


4. **The authors did not submit their code.**

5. Lastly, as a reader, I am more interested in the impact of multi-tasking on diversity strategies. For example, as mentioned in [3], the goal is for strategies to be as similar as possible across different tasks, which is contrary to the diversity score. Have the authors found similar results, and how did they address this?

[1] Kumar, Saurabh, et al. "One solution is not all you need: Few-shot extrapolation via structured maxent rl." Advances in Neural Information Processing Systems 33 (2020): 8198-8210.

[2] Chen, Wentse, et al. "DGPO: discovering multiple strategies with diversity-guided policy optimization." Proceedings of the AAAI Conference on Artificial Intelligence. Vol. 38. No. 10. 2024.

[3] Teh, Yee, et al. "Distral: Robust multitask reinforcement learning." Advances in neural information processing systems30 (2017).

**Questions:**

All my questions are mentioned in the weaknesses section.

**Limitations:**

Thoroughly discussed in Conclusions.

---

> ### Author Rebuttal · Authors · 2024-08-07
>
> We appreciate the feedback you provided on our submission.
>
>
> **In 4.1.1, the authors …**
>
> Please note that our results conform to the literature [2, 3, DOMiNO] when evaluating the Lagragian-constrained optimization in canonical MuJoCo environments and we found it to outperform the non-lagrangian counterparts. However, when considering a complex domain like Gran Turismo, where the difference between a competitive and an unsatisfactory policy could be a few hundred milliseconds (when evaluating the lap time in completing a track) and state transitions obey complex dynamic interactions, we found that Lagrangian multipliers are not as beneficial. Also, note that this kind of domain is not considered in [1, 2] or in DOMiNO.
> In complex domains, we report that Lagragian-constrained optimization fails to preserve a good performance vs diversity trade-off. This is evident in Figure (2, a) where DOMiNO is only optimizing for performance, disregarding the diversity component of the training objective. Mathematically, Lagragian-constrained optimization is known to be unstable [4] while searching for a good compromise across different constraints.
> When evaluating Lagragian-constrained optimization in DUPLEX, we found that rapid changes in the multipliers are not beneficial when learning complex environmental dynamics and the agent would either optimize for performance or diversity. This observation led us to introduce the soft-lower bound in DUPLEX – which as we report in Figure (2, b-c) outperforms other baselines.
>
> **In the experimental section...**
>
> We welcome the reviewer's suggestion and we are working towards including the diversity score of a vanilla SAC baseline in our experimental section. It is worth noting that we are normalizing diversity scores with respect to the most diverse set of policies across domains to facilitate comparison. The reported increment is then a squashed score but, as we report in Figure 3., it is important to note that DUPLEX can provide a more stable performance vs. diversity trade-off.  While we cannot provide the requested baseline yet, we can share data demonstrating the correctness of the DOMiNO implementation:
>
> - In Figure 3, the average reward and mean diversity of DOMiNO are presented for two different optimality ratios (OR). With an OR of 0.9, DOMiNO achieves an average reward of 6140 ± 94.34 and a mean diversity of 0.0215 ± 0.0086. At an OR of 0.5, the average reward is 5568 ± 698.70, with a mean diversity of 0.0372 ± 0.029. For comparison, we evaluated USFA in a single-task Walker scenario. USFA, which uses SAC and lacks diversity incentives, achieved an average reward of 6280 ± 555 and a mean diversity of 0.003 ± 0.001. USFA results are also based on 5 independent runs. These findings indicate that DOMiNO produces policies that are 7 to 12 times more diverse than those of USFA. The plot normalizes these differences, which appear minimal when compared with DUPLEX.
> - In our code-base, both DOMiNO and DUPLEX use the same utility functions; core computation of the intrinsics reward; Lagrange-constrained optimization, and SAC as the base algorithm.
> - Figure 4 illustrates the performance of different methods on a multi-context benchmark. As mentioned in Section 4.2, with infinite environments, we cannot use the average of inputs to compute Successor Features (SFs) and must estimate them instead. This introduces instability, causing both SFs and diversity rewards to chase correlated errors. DOMiNO lacks the stabilizing mechanisms introduced with DUPLEX, resulting in its failure to learn competitive and diverse policies, similar to USFA. Figure 5 highlights the importance of these mechanisms, showing that without them, DUPLEX would also fail. Appendix B demonstrates that incorporating entropy regularization into USFA (as in DUPLEX) significantly improves its results.
>
> **Minor corrections**
>
> Thank you for the pointer, we integrated these corrections to the main body of the paper.
>
> **Code.**
>
> We cannot share the code to this date. We included a detailed pseudo-code description of our algorithm, described our environment domains, and included the hyper-parameters used for training DUPLEX agents. We believe this to be the key information needed to reproduce our experiments but we are happy to include further details otherwise.
>
> **Lastly, as a reader…**
>
> The approach presented in [3] exploits policy distillation to extrapolate common skills from strategies learned across the different tasks. As pointed out by the reviewer, the diversity score instead serves a different purpose. That is, in contrast to [3], our training objective is based on encouraging diversity subject to the performance of the target policy. Such diversity is especially important in generalizing to OOD tasks, as illustrated in Figure 4 b). This is because policies can converge to strategies that share fundamental skills but learn different strategies that can be useful to unseen tasks. For example, in our Walker2D environment with changing gravity, a policy that applies higher torques to the joints has a higher chance to succeed under higher gravitational force conditions.
>
> Finally, demonstrating near-optimal performance while combining diversity and multitask generalization has direct applications to the GT player community, increasing replayability and engagement for users by racing against diverse and competitive opponents. Being robust towards OOD tasks and environments makes these agents more applicable to new game functionalities and updates.
>
> Thank you for your feedback. Let us know if you have any further questions.
>
> [4] Moskovitz, Ted, et al. "Reload: Reinforcement learning with optimistic ascent-descent for last-iterate convergence in constrained mdps." International Conference on Machine Learning. PMLR, 2023.

---

> > ### Comment · Reviewer_zFsC · 2024-08-09
> >
> > Thank you for the authors' response. However, the following issues remain unresolved during the rebuttal period:
> >
> > 1. How does multiplying the reward function by lambda affect the objective function mathematically?
> >
> > 2. Results for the vanilla SAC baseline (with lower rewards) have not been provided.
> >
> > 3. (minor) [3] suggests that different networks corresponding to multiple tasks should be as similar as possible, while the goal of DUPLEX is to make these networks perform as differently as possible. These objectives seem contradictory. How do the authors address this?

---

> > > ### Author Response · Authors · 2024-08-12
> > > **Follow up**
> > >
> > > Thank you for your follow up. Please, find our answers below:
> > >
> > > **How does multiplying the reward function by lambda affect the objective function mathematically?**
> > >
> > > Lambda is a non-linear continuous function designed to zero-out the diversity reward component of policies not in the objective region. At the same time lamba is not affecting policies that are successful in solving the target task while searching for diversity.
> > >
> > > Specifically, let us consider the case in which we set $\beta$ to 0.8 as the desired value. In this case, we are configuring the training objective to disregard policies that get an average reward less than 0.8 times the reward of the target policy, the target policy only optimizes for the extrinsic rewards. Then, Lambda is a vector that modulates the contribution of the intrinsics reward for each policy in the set. In fact, it downweights the intrinsic rewards for policies that get an average reward less than $\beta$ times the rewards of the target policy and converges to 1 when the policy gets rewards above the same threshold.
> > > Lambda is computed as a sigmoid and, consequently, does not act as a hard limit. That is, Lambda can be different from 0 or 1 with values slightly lower or higher than  $\beta$ times the target policy. Nevertheless, making this limit - and consequently the rewards - continuous, provided the additional stability that we were missing from Lagrange-constrained optimization for DUPLEX to learn competitive diverse behaviors in the GranTurismo.
> > >
> > > **Results for the vanilla SAC baseline (with lower rewards) have not been provided.**
> > >
> > > We are working towards adding such a baseline to the paper. Nevertheless, even through we realize it cannot replace the requested baseline, as a temporary answer, we provided additional details on our implementation of DOMiNO, and also highlighted that since DUPLEX is built on the foundations of DOMiNO, all the code from DOMiNO is used in DUPLEX. That includes the underlying SAC, the diversity function, the lagrange-constrained optimization and the neural network.
> > >
> > > The differences in code were the novel mechanisms introduced in Section 4.1.1 and 4.2, including the regularization term in the objective function to estimate the SFs.
> > >
> > >
> > > **(minor) [3] suggests that different networks corresponding to multiple tasks should be as similar as possible, while the goal of DUPLEX is to make these networks perform as differently as possible. These objectives seem contradictory. How do the authors address this?**
> > >
> > > We want to highlight that our objectives are not contradictory, and actually they have some resemblance. In [3] authors state that they have N policies learning to solve N tasks, where the tasks have some elements in common. In [3], authors found that encouraging the N policies to behave similarly across different tasks was beneficial.
> > >
> > > By following the notation used in [3] where the authors label a policy that learns a task as a new policy. In DUPLEX, we have M policies that solve N tasks, and according to [3] we are training M*N policies. In our notation, instead, a policy learning a task is not considered a new policy
> > >
> > > This distinction is important because it allows us to see that for every policy of the same N group, DUPLEX does not encourage them to be different between them. DUPLEX encourages diversity to the closest variant of policies from another N group.
> > >
> > > From here we can extract a first conclusion that DUPLEX and [3] are not contradictory. Moreover, since in our case N is infinite and is provided as input (ie. the context vector) we are encouraging the M policies by design to be similar -this is a property of Universal Estimator frameworks [4]. A similarity which, to a certain extent, can be further incentivised by the diversity rewards, that discourages any policy from the N group to differ from the others in a direction that is already explored in another group of N policies.
> > >
> > > [4] Borsa, Diana, et al. "Universal Successor Features Approximators." International Conference on Learning Representations.

---

> > > > ### Comment · Reviewer_zFsC · 2024-08-14
> > > >
> > > > Thank you for addressing most of my concerns. I have raised my score to 5. We recommend that the authors work on improving the paper's writing, as highlighted in our initial review, and include an explanation of the motivation behind the design of the heuristic lambda in the next version of the paper to enhance clarity.

---

### Author Rebuttal · Authors · 2024-08-07

Thank you for the detailed feedback and insightful comments. We appreciate the time and effort you have invested in this review which has significantly strengthened our contribution.
We considered each point highlighted by the individual reviewers and made several revisions to address their concerns. Here is an overview of the major changes:

- **Clarity**: We have revised the paper to address all the individual concerns raised by the reviewers. These changes have enhanced the overall presentation and made the paper clearer.
- **Related Work and Discussions**: We added the suggested citations and expanded our discussion on limitations and future work for DUPLEX.
- **Canonical RL baselines**: We have highlighted the differences with respect to non-diversity-optimizing baselines and provided additional data points in standard environments.
- **Diversity Rewards**: We have emphasized that the core computation of the diversity rewards for both DOMiNO and DUPLEX are the same.
- **Universal Estimators**: We have provided a clearer explanation of the motivation behind universal estimators in Section 4.1 to enhance understanding and clarity.
- **Code**: Unfortunately, we are unable to share the implementation at this time. The code includes proprietary content and sharing it will infringe our copyright.
- **DOMiNO Visualizations**: we are working towards including additional visualizations of DOMiNO to illustrate its performance and behavior more effectively. In the mean time, we added more data-points and insights into the answers to the rewards.

Thank you once again for your valuable feedback. We are confident that these revisions have strengthened our paper.

---

### Decision · Program_Chairs · 2024-09-25

**Decision:**

Accept (poster)

**Comment:**

Overall, reviewers were mildly positive about this paper, with several thinking the topic is important. Most seemed satisfied with the experimental evaluation. They mentioned "detailed ablations ... is sufficient", the "behavior in GranTurismo 7 is interesting and it's indeed diverse yet competent", and "the experiments performed are comprehensive". Overall, the contribution seems solid enough and worth accepting. Below, I expand on my reasoning below by summarizing the final opinions and justifications of the reviewers.

Reviewer zFsC's concerns were addressed, although they did not acknowledge that their final 3 concerns were addressed. It appears that their remaining concern 2. about needing a vanilla SAC baseline was not addressed. The authors mentioned that they are working on it but that it is not complete. This concern stemmed mainly from a doubt about the integrity of the implementation of another baseline, DOMiNO. The authors clarified that their implementation is built on top of DOMiNO, which lends more credibility to the integrity of its implementation.

Reviewer Eqtv did not give us an update after the author response. After reading their concerns and author response again, the concerns seem plausibly resolved, except for the concern about the absence of the implementation code.

Reviewer 9gQo considers the paper complete and sound, but had concerns about the weakness of the claims and extent of the novelty. The authors responded to these points, and I found their responses reasonable, adding support for the extent of the novelty.

Reviewer VFg8 did not have any major concerns (they recommended weak accept), and stated that they maintain this score after reading the author response.

Reviewer XDei recommended borderline accept, and stated their major concerns were answered/mitigated by the author response.